# An Optimization Design of Adaptive Cruise Control System Based on MPC and ADRC

Zengfu Yang [1], Zengcai Wang [2,3,*] and Ming Yan [2]

1 Shendong Coal Technology Research Institute of National Energy Group, Beijing 719315, China; yzfyjs2006@126.com
2 School of Mechanical Engineering, Shandong University, No. 17923 Jingshi Road, Jinan 250061, China; 201820385@mail.sdu.edu.cn
3 Key Laboratory of High Efficiency and Clean Mechanical Manufacture, Ministry of Education, Shandong University, Jinan 250061, China
* Correspondence: wangzc@sdu.edu.cn

**Abstract:** In this paper, a novel adaptive cruise control (ACC) algorithm based on model predictive control (MPC) and active disturbance rejection control (ADRC) is proposed. This paper uses an MPC algorithm for the upper controller of the ACC system. Through comprehensive considerations, the upper controller will output desired acceleration to the lower controller. In addition, to increase the accuracy of the predictive model in the MPC controller and to address fluctuations in the vehicle's acceleration, an MPC aided by predictive estimation of acceleration is proposed. Due to the uncertainties of vehicle parameters and the road environment, it is difficult to establish an accurate vehicle dynamic model for the lower-level controller to control the throttle and brake actuators. Therefore, feed-forward control based on a vehicle dynamic model (VDM) and compensatory control based on ADRC is used to enhance the control precision and to suppress the influence of internal or external disturbance. Finally, the proposed optimal design of the ACC system was validated in road tests. The results show that ACC with APE can accurately control the tracking of the host vehicle with less acceleration fluctuation than that of the traditional ACC controller. Moreover, when the mass of the vehicle and the slope of the road is changed, the ACC–APE–ADRC controller is still able to control the vehicle to quickly and accurately track the desired acceleration.

**Keywords:** adaptive cruise control; hierarchical control mode; acceleration predictive estimation aided MPC; active disturbance rejection control





## 1. Introduction

The adaptive cruise control (ACC) system is an extension of the traditional cruise control system. It can replace the driver to operate the accelerator and brake pedals to control a vehicle's speed and acceleration in many scenarios, including in traffic jams, and to maintain a reasonable distance or speed relative to the front target vehicle. The use of this function can greatly reduce the driver's workload and can improve the convenience of the vehicle [1–4]. Due to the development of technology and the reduction in the cost of related equipment, an increasing number of vehicles will be equipped with this technology to improve the functionality of vehicles [5–9].

To better achieve the functioning of the ACC system, scholars typically use a hierarchical control structure, including an upper and lower controller [10]. The upper-level controller uses the perception layer data and vehicle parameters as controller inputs to determine the typical desired control command for a vehicle with ACC, whereas the lower-level controller controls the throttle and brake actuators to track the desired acceleration [11]. The upper-level controller of the ACC system acts in the place of the driver, determining how to operate the ACC vehicle. Adaptive cruise control typically has two types of control actions: velocity control and spacing control [12]. For the velocity control

mode, an upper speed limit should be preset by the driver when the ACC function is turned on. This ACC system controls the host vehicle by ensuring that it drives at the preset speed if there is no forward target vehicle in the same lane. When the radars detect that there is a vehicle ahead in the same lane, the system uses the distance control mode to maintain a reasonable distance between the host vehicle and the front vehicle. Compared with the velocity control mode, the spacing control mode faces more complicated situations.

To realize the functioning of the adaptive cruise control system, various controllers have been adopted by different scholars. Zhang et al. [13] used the classical PID controller to adjust the relative space error and relative velocity to the front target vehicle and selected appropriate control parameters through the pole-zero placement. Yi et al. [14] combined a PI model and feed-forward method to design an upper-level controller; this approach gives the system a faster response speed. Naranjo et al. [15] proposed a new global error function, which enables the use of heuristic optimization methods for PID tuning. Li et al. [16] designed a car-following algorithm based on the sliding mode control method to address the problems of the PID controller, namely, that it is prone to overshooting and requires a long time to reach a stable state. Zhang et al. [17] used a fuzzy controller to design an ACC system with a stop-and-go function, and the simulation results showed that it has a good level of robustness.

In recent years, many scholars have adopted model predictive control as the upper controller of the ACC system [18–21]. One important reason for this approach is that an MPC controller can take multiple control requirements into account at the same time, including requirements that contradict each other [22]. LUO et al. [23] noted that using the MPC controller as the ACC system's upper controller can improve EV energy consumption.

Plessen et al. [24] presented an integrated control approach for autonomous vehicles to realize longitudinal control based on the MPC method and reconstructed a vehicle model as a nonlinear dynamic bicycle model. Naus et al. [25] adopted an explicit MPC synthesis as the ACC upper controller, which is able to explicitly address the optimization problem offline rather than online. To obtain the optimal acceleration solution, the MPC prediction model must be accurate and reliable. When radars detect that there is a vehicle ahead in the same lane, the system uses the distance control mode to maintain a reasonable distance and speed between the host vehicle and the front vehicle. Hence, the information about the front vehicle, and particularly the front vehicle's acceleration, is important for the MPC controller to be able to control the host vehicle. In practice, the acceleration of the front vehicle does not remain constant when accelerating or decelerating. However, when developing predictive models of the future state in traditional MPC controller design [26–28], the acceleration of the target vehicle is always considered a fixed value. This leads to inappropriate optimal solutions under some following conditions. Therefore, an estimator to enable prediction of the acceleration of the target vehicle is needed for the MPC controller.

Another critical challenge in the ACC system is the accuracy of acceleration tracking of the lower-level controller. It is challenge to obtain an accurate vehicle model. Zhang et al. [29] reduced the dynamic model complexity to ease its use in the controller. When the MPC controller is used as the upper controller, the accelerating and braking actions are calculated by the inverse longitudinal vehicle model (ILVM) of the lower-level control [30]. The control precision is significantly influenced by internal and external disturbances, which include the non-linear vehicle dynamic model, the change in the mass of the vehicle, the varying road resistance, and the road slope. Each of these disturbances can influence the performance of acceleration tracking. In particular, it is a challenge for the ILVM to calculate the control inputs in the presence of disturbances. To resolve this issue, a compensation method is required to address the influence caused by the disturbances. In this study, the ADRC method was used to ensure the desired accuracy of acceleration tracking.

The contributions of this study can be summarized as follows: (a) To address the difficulty faced by a traditional ACC controller to optimize multiple performance objectives at the same time, model predictive control was used in this study as the upper controller

to balance the safety, tracking capability, fuel economy, and ride comfort of the ACC system. (b) To address the problem that the traditional MPC regards the acceleration of the target vehicle as a fixed value, resulting in deviation from the expected acceleration, an acceleration prediction estimator based on the least square method and the acceleration history information of the front vehicle was applied to the prediction model. (c) To address the problem of vehicle dynamic model inaccuracy caused by changing road environment and vehicle parameters, the strategy of combining ADRC feedback compensation and VDM-based feed-forward control was adopted. When the dynamic model parameters change, the lower actuator can still accurately and quickly track the output of the decision-making layer.

## 2. Materials and Methods

A hierarchical construction was used to design the ACC system, which consists of an upper-level controller and a lower-level controller, as shown in Figure 1. According to the inputs, the upper-level controller outputs an optimal desired acceleration $a_{des}$. The lower-level controller determines acceleration pedal position $\alpha$ and brake pressure $P$ to control the host vehicle to track desired acceleration $a_{des}$. The paper is organized as follows. First, a three-state longitudinal car-following model for the ACC system is presented. Second, the design of the upper-level controller based on the APE and MPC methods is discussed. Then, the design the lower-level controller using a feed-forward control and compensatory control based on the ADRC method is presented. Next, the results of three road tests undertaken to validate the proposed ACC controller are discussed. Finally, conclusions are presented.

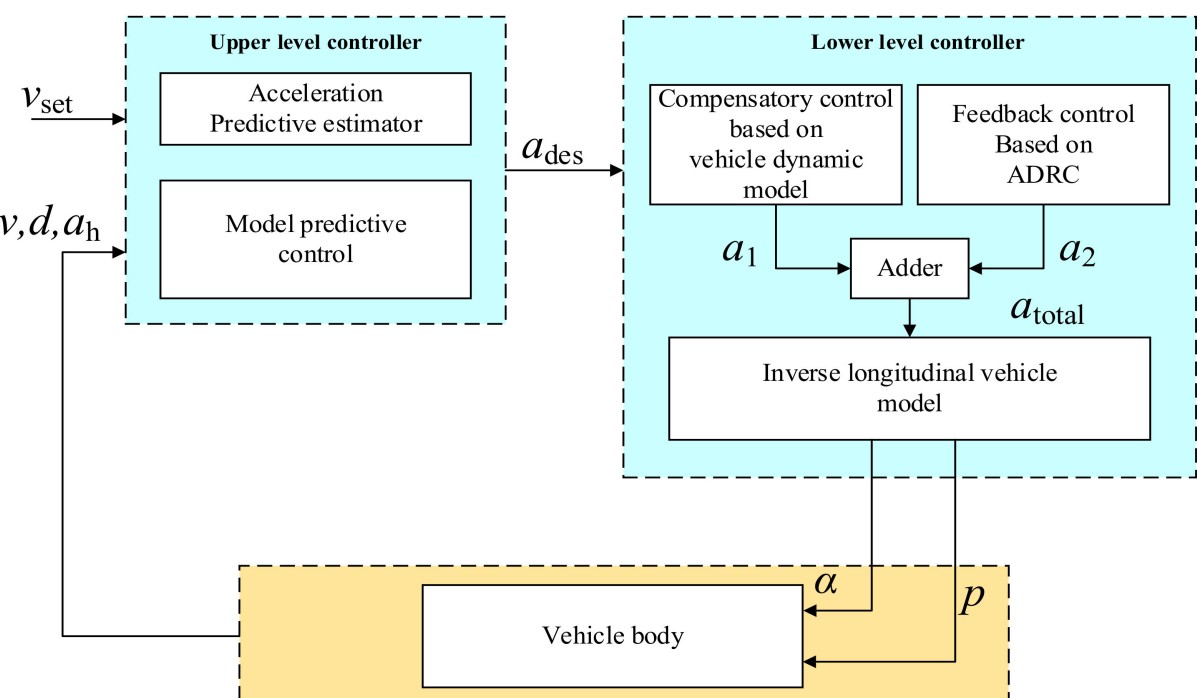

**Figure 1.** Schematic representation of the ACC system.

### 2.1. Longitudinal Car-Following Model

There are two main reasons for the strong nonlinear characteristics of the longitudinal driving of vehicles. First, the output of the engine is not linear, and the transmission ratio and the air resistance are time varying. Second, there is a time delay in the execution of actuators, including those of the accelerator and brake pedals, after receiving an input signal, despite the introduction of a first-order lag into the car-following model [31]. There

are two vehicles in the car-following model, located in the front and rear. The following vehicle located in the rear is defined as the host vehicle, and the followed vehicle in the front is defined as the target vehicle.

The relationship between the desired acceleration and the actual acceleration of the host vehicle can be defined as follows:

$$\dot{a}_h(k+1) = \frac{K_L}{T_L} a_{des}(k) - \frac{1}{T_L} a_h(k) \tag{1}$$

where $a_h$ is the actual acceleration of the host vehicle; $a_{des}$ is the desired acceleration of the host vehicle; $K_L$ is the system gain coefficient; and $T_L$ is the time coefficient. With respect to the longitudinal kinetic characteristic of the front vehicle and the host vehicle, two state variables are defined: (1) the distance error between the actual distance and the desired distance, which can be expressed as $\Delta d$; (2) the speed error between the speed of the front car and the speed of the host car, which can be expressed as $\Delta v$. According to the desired safe vehicle spacing model [25], the two state variables can be formulated as follows:

$$\begin{cases} \Delta d = d - d_{des} \\ \Delta v = v_f - v_h \\ d_{des} = t_h v_h + d_0 \end{cases} \tag{2}$$

where $d$ is the distance obtained by the radar; $d_{des}$ is the desired distance; $v_f$ is the actual speed of the target vehicle; $v_h$ is the speed of the host vehicle; $t_h$ is the constant time headway; and $d_0$ is the minimum distance when the vehicles are stopped.

Considering the tracking capability, safety, fuel economy, ride comfort, and the capacity of the on-board processor, $\Delta d$, $\Delta v$, and $a_h$ were selected as the three states of the state vector of the longitudinal vehicle model, shown as follows:

$$x = \begin{bmatrix} \Delta d & \Delta v & a_h \end{bmatrix}^T \tag{3}$$

The differential form of the state variable can be formulated as follows:

$$\begin{cases} \dot{\Delta d} = \Delta v - t_h a_h \\ \dot{\Delta v} = a_f - a_h \\ \dot{a}_h = \frac{K_L}{T_L} a_{des} - \frac{1}{T_L} a_h \end{cases} \tag{4}$$

where $a_f$ is the acceleration of the front target vehicle.

According to Equations (3) and (4), the three-state space model can be described as follows:

$$\dot{x} = \begin{bmatrix} \dot{\Delta d} \\ \dot{\Delta v} \\ \dot{a}_h \end{bmatrix} = \begin{bmatrix} 0 & 0 & -t_h \\ 0 & 0 & -1 \\ 0 & 0 & -1/T_L \end{bmatrix} \begin{bmatrix} \Delta d \\ \Delta v \\ a_h \end{bmatrix} + \begin{bmatrix} 0 \\ 0 \\ K_L/T_L \end{bmatrix} a_{des} + \begin{bmatrix} 0 \\ 1 \\ 0 \end{bmatrix} a_f \tag{5}$$

Hence, the car-following model of the upper controller can be expressed as follows:

$$\begin{cases} \dot{x} = Ax + Bu + Cw \\ y = Gx \\ u = a_{des} \\ w = a_f \end{cases} \quad A = \begin{bmatrix} 0 & 0 & -t_h \\ 0 & 0 & -1 \\ 0 & 0 & -1/T_L \end{bmatrix} \quad B = \begin{bmatrix} 0 \\ 0 \\ \frac{K_L}{T_L} \end{bmatrix} \quad C = \begin{bmatrix} 0 & 1 & 0 \end{bmatrix} \quad G = \begin{bmatrix} 1 & 0 & 0 \\ 0 & 1 & 0 \\ 0 & 0 & 1 \end{bmatrix} \tag{6}$$

where $u$ is the control input; $y$ is the output of the model; and $w$ is the disturbance of the system.

The MPC controller is calculated and realized by the computer, so the continuous-time car-following model (6) need to be converted into a discrete-time form. Due to the existence of a complex exponential matrix and its integral term, accurate discretization requires a significant computational effort. In this paper, a method called the forward Euler

(FE) method is used to accelerate the computation [32]. The detail of the method can be expressed as follows:

$$x(k+1) = (I + AT)x(k) + TBu(k) \tag{7}$$

where $T$ is the system sampling time and $I$ is a three-state identity matrix.

According to the FE method, the discrete-time form of the car-following model (7) can be expressed as follows:

$$\begin{cases} x(k+1) = A_{d,t}x(k) + B_{d,t}u(k) + C_{d,t}w(k) \\ \qquad\qquad y(k) = Gx(k) \end{cases} \tag{8}$$

where $A_{d,t}$, $B_{d,t}$, $C_{d,t}$ are the discrete system matrices, shown as follows:

$$A_{d,t} = I + AT = \begin{bmatrix} 1 & T & -Tt_h \\ 0 & 1 & -T \\ 0 & 0 & 1 - T/T_L \end{bmatrix}$$

$$B_{d,t} = TB = \begin{bmatrix} 0 & 0 & T\frac{K_L}{T_L} \end{bmatrix}^T \tag{9}$$

$$C_{d,t} = TC = \begin{bmatrix} 0 & T & 0 \end{bmatrix}^T$$

### 2.2. The Design of the Upper-Level Controller

To improve the accuracy of the predictive model in MPC, the disturbance term should be reasonably predicted and estimated. In the following process, the acceleration of the vehicle in front is not constant. In the rolling optimization process of MPC, traditional MPC takes the acceleration value of the target vehicle as a constant value, resulting in a certain deviation from the desired acceleration. Therefore, this paper proposes an acceleration prediction estimator based on the least square method and the acceleration history information of the front vehicle and applies it to the prediction model.

#### 2.2.1. The Design of the Acceleration Predictive Estimator

Based on the history information of the front vehicle's acceleration, the acceleration of the front vehicle is predicted using the least squares method. The acceleration estimator is then applied to the predictive control frame of the car-following model.

Before the estimation, the following assumption should be made: During a short period, the acceleration of the target vehicle is changed along a straight line, shown as follows:

$$a_f(t) = a_0 + a_1 t \tag{10}$$

where $a_0$ is the initial value and $a_1$ is the slope of the function. Hence, according to the value of the current time and the value of the previous period, $a_0$ and $a_1$ can be calculated by the least squares method.

According to the equation and the current sample value $a_f(k)$, the estimated value $\bar{a}_f(t)$ can be calculated as follows:

$$\bar{a}_f(t) = a_f(k) + a_1(t-k) = a_f(k) - a_1 k + a_1 t \tag{11}$$

where $k$ is the current time, $t$ is the future time.

There are $p-1$ past sample values, for which the values are $a_f(k+1-p)$, $a_f(k+2-p)$, $\cdots$, $a_f(k \Sigma 1)$.

To ensure that the straight line is approximate to the value at other sample times, a cost function is introduced:

$$J_a = \sum_{i=k+1-p}^{k-1} q_i \left( a_f(i) - \bar{a}_f(i) \right)^2 \tag{12}$$

where $q_i$ is the weighting matrix.

The equation to obtain the value of $a_1$ is then derived, shown as follows:

$$\frac{dJ_a}{da_1} = 2 \sum_{i=k+1-p}^{k-1} q_i(a_f(i) - \overline{a}_f(i))(i-k) = 0$$

$$a_1 = \frac{\sum_{i=k+1-p}^{k-1} q_i(a_f(i) - a_f(k))}{\sum_{i=k+1-p}^{k-1}(i-k)} \qquad (13)$$

According to Equation (12), the acceleration predictive sequence can be obtained as follows:

$$\begin{bmatrix} a_f(k+1) \\ a_f(k+2) \\ \vdots \\ a_f(k+p) \end{bmatrix} = \begin{bmatrix} 1 & k+1 \\ 1 & k+2 \\ \vdots & \vdots \\ 1 & k+p \end{bmatrix} \times \begin{bmatrix} a_f(k) - \dfrac{\sum_{i=k+1-p}^{k-1} q_i(a_f(i) - a_f(k))}{\sum_{i=k+1-p}^{k-1}(i-k)} \\ \dfrac{\sum_{k=k+1-p}^{k-1} q_i(a_f(i) - a_f(k))}{\sum_{i=k+1-p}^{k-1}(i-k)} \end{bmatrix} \qquad (14)$$

### 2.2.2. Design of the Model Predictive Control Controller

The analysis of the control objective is shown as follows:

- The safety objective

ACC is a driver assistance system; thus, in both dynamic and static vehicle working processes, safety is the foremost issue to be considered during development. When applying the ACC system, the host vehicle should always maintain a safe distance from the front target vehicle.

To avoid accidents, it is necessary to restrict the minimum space between the host vehicle and the front vehicle. A time-to-collision method was introduced to provide the minimum safe distance when the relative speed is not zero [33]. Based on the above, the objective of safety can be expressed as follows:

$$\begin{cases} d(k) \geq d_{\min}(k) \\ d_{\min}(k) = \max\{TTC \cdot \Delta v(k), d_0\} \end{cases} \qquad (15)$$

where $d_{\min}(k)$ is the minimum safe distance at the sample time $k$, $TTC$ is defined as the estimated time at which a collision will occur.

- The tracking capability objective

The basic goal of an adaptive cruise control system is to follow the target vehicle and maintain a desired distance from it. This objective can be divided into two situations:

1.  When the front target vehicle is driving in a stable state, the values of distance error and speed error should be controlled to zero, which can be expressed as follows:

$$\begin{cases} \Delta d(k) \to 0 \\ \Delta v(k) \to 0 \end{cases}, \qquad k \to \infty \qquad (16)$$

2.  When the front target vehicle is in the state of braking or accelerating, the distance between the two vehicles should be restricted to prevent a cut-in, which can be expressed as follows:

$$\begin{cases} \Delta d_{\min} \leq \Delta d(k) \leq \Delta d_{\max} \\ \Delta v_{\min} \leq \Delta v(k) \leq \Delta v_{\max} \end{cases} \qquad (17)$$

- The fuel economy objective

When the host vehicle is following the target vehicle, the host vehicle must accelerate or brake to maintain the desired distance. The smoother the driving behavior, the better the fuel economy [34]. Therefore, the fuel economy can be evaluated according to the value of the acceleration and the control output. To achieve good fuel economy, the value of acceleration and control output should be restricted, and can be described as follows:

$$\begin{cases} a_{\min} \leq a_h(k) \leq a_{\max} \\ u_{\min} \leq u(k) \leq u_{\max} \end{cases} \tag{18}$$

where $a_{\min}$, $a_{\max}$, $u_{\min}$, and $u_{\max}$ are the upper and lower limits of the acceleration and the control outputs, respectively.

- The ride comfort objective

To improve the riding comfort of the driver and passengers, the jerk $j(k)$ of the vehicle should be reduced when the vehicle follows the front target vehicle. Jerk is a derivative of acceleration, which reflects the change rate of acceleration. The change rate of control quantity should be limited. This can be described as follows:

$$\begin{cases} j_{\min} \leq j(k) \leq j_{\max} \\ \Delta u_{\min} \leq \Delta u(k) \leq \Delta u_{\max} \end{cases} \tag{19}$$

where $j_{\min}$, $j_{\max}$, $\Delta u_{\min}$, and $\Delta u_{\max}$ are the upper and lower limits of the acceleration change rate and the control output change rate, respectively.

According to the principle of MPC, the state of the host vehicle in the predicted future horizon can be predicted using the car-following model and the current state at each sampling time. By solving an optimization problem, a control sequence can be acquired, and the first control action is chosen as the control output for the system. The above steps are repeated at the next sampling time. To reduce or eliminate the static error, Equation (8) is changed into an incremental form, which can be expressed as follows:

$$\begin{cases} \Delta x(k+1) = \boldsymbol{A}_{d,t}\Delta x(k) + \boldsymbol{B}_{d,t}\Delta u(k) + \boldsymbol{C}_{d,t}\Delta w(k) \\ y(k) = \boldsymbol{G}\Delta x(k) + y(k-1) \end{cases} \tag{20}$$

where $\Delta x$ is the increment of state variables, $\Delta u$ is the increment of control variables, and $\Delta w$ is the increment of acceleration disturbance.

$\Delta x$, $\Delta u$, and $\Delta w$ can be described as follows:

$$\begin{cases} \Delta x(k) = x(k) - x(k-1) \\ \Delta u(k) = u(k) - u(k-1) \\ \Delta w(k) = w(k) - w(k-1) \end{cases} \tag{21}$$

To derive the prediction equation of the car-following system, the following assumptions must be made: when the time of the predicted horizon is beyond the control horizon, the values of the control output remain constant so that $\Delta u(k+i) = 0, i = m, m+1, \ldots, p-1$, where m is the control time domain and $p$ is the prediction time domain.

At sampling time $k$, $\Delta x(k)$ is taken as the starting point to predict the future dynamics of the system. From the above equation, the state from $k$ to $k+p-1$ and the predicted output from $k+1$ to $k+p$ can be predicted. The predictive output matrix of the system in the predictive time domain is represented as follows:

$$\boldsymbol{Y}_P(k+1|k) = \boldsymbol{S}_x\Delta x(k) + \boldsymbol{I}y(k) + \boldsymbol{S}_u\Delta \boldsymbol{U}(k) + \boldsymbol{S}_w\Delta \boldsymbol{W}(k) \tag{22}$$

where $\mathbf{S}_x$, $\mathbf{I}$, $\mathbf{S}_w$, and $\mathbf{S}_u$ are the coefficient matrices. The vectors of the control inputs, system disturbances, and predicted outputs can be expressed as follows:

$$\Delta\mathbf{U}(k) \stackrel{def}{=} \begin{bmatrix} \Delta u(k) \\ \Delta u(k+1) \\ \vdots \\ \Delta u(k+m-1) \end{bmatrix}_{m\times 1}$$

$$\Delta\mathbf{W}(k) \stackrel{def}{=} \begin{bmatrix} 1 & k \\ 1 & k+1 \\ \vdots & \vdots \\ 1 & k+p-1 \end{bmatrix}_{p\times 2} \begin{bmatrix} a_p(k) - \dfrac{\sum\limits_{i=k+1-p}^{k-1} q_i \begin{pmatrix} a_p(i) \\ -a_p(k) \end{pmatrix}}{\sum\limits_{i=k+1-p}^{k-1}(i-k)} \\ \dfrac{\sum\limits_{i=k+1-p}^{k-1} q_i \begin{pmatrix} a_p(i) \\ -a_p(k) \end{pmatrix}}{\sum\limits_{i=k+1-p}^{k-1}(i-k)} \end{bmatrix} \tag{23}$$

$$\mathbf{Y}_p(k+1|k) \stackrel{def}{=} \begin{bmatrix} y(k+1|k) \\ y(k+2|k) \\ \vdots \\ y(k+p|k) \end{bmatrix}_{p\times 1}$$

To satisfy the requirements for safety, improved fuel economy, tracking capability, and passenger ride comfort, the optimal control output based on the car-following model should be obtained by minimizing the objective function, which can be expressed as follows:

$$J(x, \Delta u, m, p) = \left\| \mathbf{\Gamma}_y \big( \mathbf{Y}_p(k+1|k) - \mathbf{R}(k+1) \big) \right\|^2 + \left\| \mathbf{\Gamma}_u \Delta\mathbf{U}(k) \right\|^2 \tag{24}$$

where $\mathbf{\Gamma}_y$ and $\mathbf{\Gamma}_u$ are the weighting matrices of the predicted output and system control input, respectively, and $\mathbf{R}(k+1)$ is the reference input vector of the system at the sampling time $k+1$.

$$\begin{cases} \mathbf{\Gamma}_y = diag\big(\Gamma_{yp}(k), \Gamma_{yp}(k), \cdots, \Gamma_{yp}(k)\big)_{p\cdot p} \\ \mathbf{\Gamma}_u = diag\big(\Gamma_{um}(k), \Gamma_{um}(k), \cdots, \Gamma_{um}(k)\big)_{m\cdot m} \end{cases}$$

$$\mathbf{R}(k+1) = \begin{bmatrix} r(k+1) \\ r(k+2) \\ \vdots \\ r(k+p) \end{bmatrix}_{p\times 1} \tag{25}$$

where $r(k+i), i = 1, 2, \ldots . p$ is the reference trajectory of the car-following model.

Based on the above method, the cost function of the multiple objective car-following model can be expressed as follows:

$$\min_{\Delta U(K)} J(\mathbf{Y}_p(k), \Delta\mathbf{U}(k), m, p)$$
$$s.t. \ (16) \quad (17) \quad (18) \quad (19) \tag{26}$$

Due to the existence of system constraints, a numerical optimization method is needed. To transform Equation (24) into a form of a quadratic programming model, a vector is introduced, which is irrelevant to the system control input. This can be described as follows:

$$\mathbf{E}_p(k+1|k) = \mathbf{R}(k+1) - \mathbf{S}_x \Delta x(k) - Iy(k) - \mathbf{S}_w \Delta\mathbf{W}(k) \tag{27}$$

Combining Equations (22), (26) and (27), the cost function of the car-following problem is transformed as:

$$\min_{\Delta U(K)} \Delta\mathbf{U}(k)^T \mathbf{H} \Delta\mathbf{U}(k)^T - G(k+1|k)^T \Delta\mathbf{U}(k)$$
$$s.t. \quad C_u \Delta\mathbf{U}(k) \geq \mathbf{b}(k+1|k) \tag{28}$$

where $\mathbf{H} = \mathbf{S}_u^T \boldsymbol{\Gamma}_y^T \boldsymbol{\Gamma}_y \mathbf{S}_u + \boldsymbol{\Gamma}_u^T \boldsymbol{\Gamma}_u$ and $\mathbf{G}(k+1|k) = 2\mathbf{S}_u^T \boldsymbol{\Gamma}_y^T \boldsymbol{\Gamma}_y \mathbf{E}_p$ are the parameter matrices, and $\boldsymbol{C}_u$ and $\mathbf{b}(k+1|k)$ are the matrices that are relevant to the constraints, which can be expressed as follows:

$$C_u = \begin{bmatrix} \mathbf{S}_u^T & -\mathbf{S}_u^T & \mathbf{L}^T & -\mathbf{L}^T & \mathbf{T}^T & -\mathbf{T}^T \end{bmatrix}$$

$$\mathbf{b}(k+1|k) = \begin{bmatrix} \mathbf{Y}_p(k+1|k) - \mathbf{Y}_{\min}(k+1) \\ \mathbf{Y}_p(k+1|k) + \mathbf{Y}_{\max}(k+1) \\ -\mathbf{U}_{\min} \\ \mathbf{U}_{\max} \\ \Delta\mathbf{U}_{\max} \\ -\Delta\mathbf{U}_{\max} \end{bmatrix} \tag{29}$$

The control input sequence $\Delta\mathbf{U}^*(k)$ can be calculated based on the problem (28), and the first value of the control input sequence can be chosen as the incremental control input $\Delta u(k)$, which can be expressed as follows:

$$u(k) = \begin{bmatrix} I & 0 & \cdots & 0 \end{bmatrix}_{1 \times m} \Delta\mathbf{U}^*(k) \tag{30}$$

Then, the incremental control input $\Delta u(k)$ is entered into the system to obtain the real control input, which can be described as follows:

$$u(k) = u(k-1) + \Delta u(k) \tag{31}$$

At the next sampling time, the new vehicle state $\mathbf{x}(k)$ is input into the car-following model to calculate the next optimal control input, etc.

### 2.3. The Design of the Lower-Level Controller

As shown in Figure 2, the control strategy of the lower-level controller designed in this study includes two parts: feed-forward control based on inverse longitudinal dynamics; and feedback compensation control based on ADRC. The feedback compensation control includes three parts, as shown in the dotted box in Figure 2; namely, the tracking differentiator, extended state observer, and nonlinear error feedback. Since the vehicle experiences a certain delay in the process of executing the control quantity, a first-order delay link is added before the ADRC feedback compensation control. By adding output $u$ of ADRC and the output of the feed-forward control, the sum of the two outputs is converted into the opening of the throttle or brake actuator and applied to the controlled vehicle.

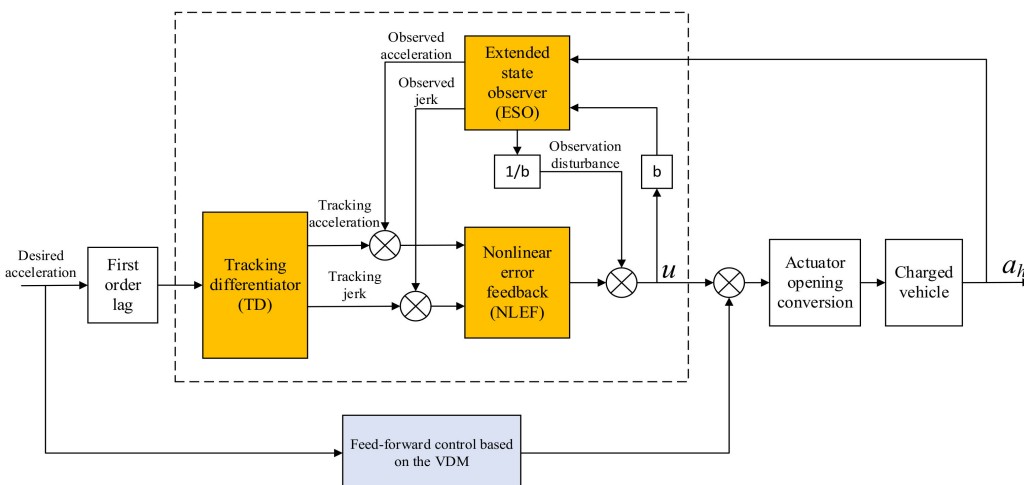

**Figure 2.** Principle of the lower-level controller.

2.3.1. Feed-Forward Control Based on the VDM

- Driving dynamic model

It is known from vehicle dynamics that the torque output from the engine is the driving power source of the vehicle. In addition, the vehicle experiences different kinds of resistance during the driving process, including rolling resistance, wind resistance, ramp resistance, and braking force exerted by the braking system [35]. Thus, the longitudinal kinematic equation can be expressed as follows:

$$ma_{des} = \frac{\tau\left(\frac{\omega_t}{\omega_e}\right)i_g i_m}{R}\eta T_{engine} - F_b - mgf\cos\theta - \frac{1}{2}C_D A\rho_A v^2 - mg\sin\theta \qquad (32)$$

where $m$ is the mass of the vehicle, $\tau$ is the speed–torque ratio coefficient, $R$ is the radius of the tire, $\omega_t$ is the turbine blade speed, $\omega_e$ is the engine speed, $i_g$ is the transmission ratio, $i_m$ is the axle main reduction ratio, $F_b$ is the braking force, $f$ is the driving resistance coefficient, $C_D$ is the wind resistance coefficient, $A$ is the cross-sectional area of the car, $\rho_A$ is the air density, and $T_{engine}$ is the engine torque. When the desired acceleration is positive, according to the driving speed, transmission gear, and engine speed, the desired engine torque can be calculated as follows:

$$T_{engine} = \frac{\left|ma_{des} + mgf\cos\theta + \frac{1}{2}C_D A\rho_A v^2 + mg\sin\theta\right|R}{\tau\left(\frac{\omega_t}{\omega_e}\right)i_g i_m \eta} \qquad (33)$$

Hence, the throttle angle can be calculated by looking up the inverse map, shown as follows:

$$\alpha_{des} = f^{-1}(T_{engine}, \omega_e) \qquad (34)$$

- Braking dynamic model

When the desired acceleration is negative, the brake force can be calculated according to the following equation:

$$F_b = -ma_{des} - mgf\cos\theta - \frac{1}{2}C_D A\rho_A v^2 - mg\sin\theta \qquad (35)$$

The desired brake pressure can be calculated as follows:

$$P_{des} = \frac{\left|-ma_{des} - \frac{1}{2}C_D A\rho_A v^2 - mg\sin\theta - mgf\cos\theta\right|}{k_b} \qquad (36)$$

where $k_b$ is the linear proportional coefficient of the brake cylinder, shown as follows:

$$K_b = \frac{T_{Fb} + T_{Rb}}{P_b R} \qquad (37)$$

where $T_{Fb}$ is the proportional coefficient between the wheel and the brake cylinder pressure of the front axle, and $T_{Rb}$ is the proportional coefficient between the wheel and the brake cylinder pressure of the rear axle.

The feed-forward control based on the VDM requires an accurate dynamic model. A change in the driving environment or the own state of the vehicle will have a significant impact on the performance of feed-forward control. For example, a change in the number of passengers leads to a change in the total mass of the vehicle, which leads to a change in the wheel radius. When the vehicle is driving on an uphill or downhill road, because the setting of the slope in the formula is also a fixed value, the control quantity used by the lower layer on the throttle or brake cannot meet the following performance of the ACC system. This reduces the comprehensive performance of the ACC system and results in security risks.

2.3.2. Compensatory Control Based on ADRC

This study adopted a strategy based on the VDM as the feed-forward control of the actuator in the lower controller. Some parameters in the model were set as fixed values in the design process, and the performance of feed-forward control is significantly reduced if the parameters of the VDM change. ADRC control is used to compensate and feedback the output of the lower level of the ACC system, so that the controller can stably and accurately track the desired acceleration of the upper-layer controller. Compared with the feed-forward control based on the VDM, ADRC does not rely on an accurate vehicle dynamics model. When vehicle parameters or the road environment changes, ADRC can make accurate changes and can improve the anti-interference ability of the whole ACC system.

The ADRC is composed of three different parts, namely, tracking differentiator (TD), nonlinear error feedback control (NLEF), and extended state observer (ESO) [36]. A second-order ADRC was designed for the lower-level controller to compensate for and suppress the control error caused by internal and external disturbances. The desired acceleration $a_{des}$ of the host vehicle is the input of the TD, and the output of the NLEF is the compensation value of the desired acceleration.

The process of the TD can be expressed as follows:

$$\begin{cases} a_1(k+1) = a_1(k) + T \cdot a_2(k) \\ a_2(k+1) = a_2(k) + T \cdot fst(a_1(k) - a_{des}(k), a_1(k), \gamma, \varpi) \end{cases} \tag{38}$$

where $a_1$ is the track value of $a_{des}$; $a_2$ is the track value of $\dot{a}_{des}$; $\gamma$ is the speed factor that determines the tracking speed; $\varpi$ is the filter factor; $fst(\cdot)$ is the simplified form of the rapid synthesis control function, shown as follows:

$$\begin{cases} d = \gamma \varpi^2 \\ \delta_0 = \varpi^2 x_2 \\ y = x_1 + \delta_0 \\ \delta_1 = \sqrt{d(d+8|y|)} \\ \delta_2 = \delta_0 + sign(y)(a_1 - d)/2 \\ S_y = (sign(y+d) - sign(y-d))/2 \\ \delta = (\delta_0 + y - \delta_2)S_y + \delta_2 \\ S_\delta = (sign(\delta+d) - sign(\delta-d))/2 \\ fst = -\gamma(a/d - sign(a))S_\delta - \gamma sign(\delta) \end{cases} \tag{39}$$

ESO redefines all uncertain parameters of the system as new state variables and then estimates them, including the system model error, external disturbance, and internal disturbance. ESO also provides error compensation for the controlled object to replace the integral feedback link in the traditional controller. The advantage of ESO is that it does not need the controlled object to provide an accurate mathematical model as the research basis and only needs to know the output and input signals of the controlled object to estimate the state variables of the system and the action on the system.

By inputting the control variable $u$ acting on the vehicle model and the actual acceleration $a_h$ output from the vehicle model into the ESO, the real-time estimated differential values $z_1$ and $z_2$ of the vehicle model and the estimated disturbance value $z_3$ caused by internal and external disturbances can be obtained simultaneously. The calculation process of ESO is:

$$\begin{cases} e = z_1(k) - a_{real}(k) \\ z_1(k+1) = z_1(k) + T(z_2(k) - \beta_1 e) \\ z_2(k+1) = z_2(k) + T(z_3(k) - \beta_2 fal(e, \alpha, \varepsilon) + bu) \\ z_3(k+1) = z_3(k) - T\beta_3 fal(e, \alpha, \varepsilon) \end{cases} \tag{40}$$

where $e$ is the error between real acceleration $a_{real}$ and control input $z_1$; $\beta_1$, $\beta_2$, and $\beta_3$ are the nonlinear parameters; $fal(\cdot)$ is the nonlinear function, which can be expressed as follows:

$$fal(e,\alpha,\varepsilon) = \begin{cases} \frac{e}{\varepsilon^{\alpha-1}}, |e| \leq \varepsilon \\ |e|^{\alpha} sign(e), |e| > \varepsilon \end{cases} \tag{41}$$

In the classical PID control, each control variable is simply combined by linear weighting. This control form is simple and effective but cannot easily address the problem of overshooting in the process of fast control. Therefore, in ADRC control, the state error feedback is described in nonlinear form, which includes the difference between TD output and ESO output, and the integral value of tracking value deviation. The process of the NLEF can be expressed as follows:

$$\begin{cases} \xi_1 = a_1(k) - z_1(k) \\ \xi_2 = a_2(k) - z_2(k) \\ u_0 = k_1 fal(\xi_1, \alpha_1, \varepsilon) + k_2 fal(\xi_2, \alpha_2, \varepsilon) \\ u = u_0 - \frac{z_3}{b} \end{cases} \tag{42}$$

where $k_1$ and $k_2$ are the weighting coefficients, $\xi_1$ and $\xi_2$ are the errors between ESO and TD, $u_0$ is the output of NLEF, $u$ is the output of the ADRC controller, and $z_3/b$ is the total compensation of internal and external disturbances.

### 2.3.3. Feed-Forward and ADRC Compensatory Control of the Lower Level

The comprehensive output acceleration of the lower-level controller is provided by the sum of the desired acceleration $a_{des}$ of the feed-forward control and the output $u$ of the ADRC controller.

- Driving control:

$$\begin{cases} T_{ADRC} = \frac{mu}{k_\psi} \\ T_{drive} = T_{des} + T_{ADRC} \end{cases} \tag{43}$$

The throttle angle can be calculated by looking up the inverse map, shown as follows:

$$\alpha_{brake} = f^{-1}(T_{drive}, \omega_e) \tag{44}$$

where $\alpha_{brake}$ is the throttle angle after the compensation of the ADRC controller.

- Braking control:

$$\begin{cases} a_{brake} = a_{des} + u \\ P_{brake} = \dfrac{\left| \begin{matrix} -ma_{brake} - \frac{1}{2}C_D A \rho_A v^2 \\ -mg\sin\theta - mgf\cos\theta \end{matrix} \right|}{K_b} \end{cases} \tag{45}$$

where $P_{brake}$ is the throttle angle after the compensation of the ADRC controller.

## 3. Road Testing Results and Discussion

To verify the performance of the proposed ACC algorithm, we used an experimental vehicle (Toyota Yaris) with a gasoline engine and automatic transmission. The hardware implementation of the test vehicle is shown in Figure 3. The perception layer includes a millimeter wave radar (MMW) and two lasers. A six-axis acceleration sensor and wheel speed sensor were also equipped to provide $a_h$ and $v_h$. The electronic throttle protocol was decrypted to allow us to control throttle position $\alpha$ based on $\alpha_{drive}$ and the inverse longitudinal vehicle model. The braking system in this vehicle was not authorized by the carmaker, so an electronic hydraulic braking system was designed. All of the data from the sensors and the electronic subsystems were connected to the data collector and an industrial personal computer through a controller area network (CAN). The signal

processing algorithm and the control strategies were programmed using the C language. For the comparison of different controllers in the identical test scenario, replicable speed and distance profiles are required. Therefore, we introduced virtual vehicles instead of genuine front vehicles in the experiment. The road tests were divided into two parts: (1) testing the differences between the ACC system, with and without APE, and (2) testing the differences between the ACC system, with and without ADRC feedback control.

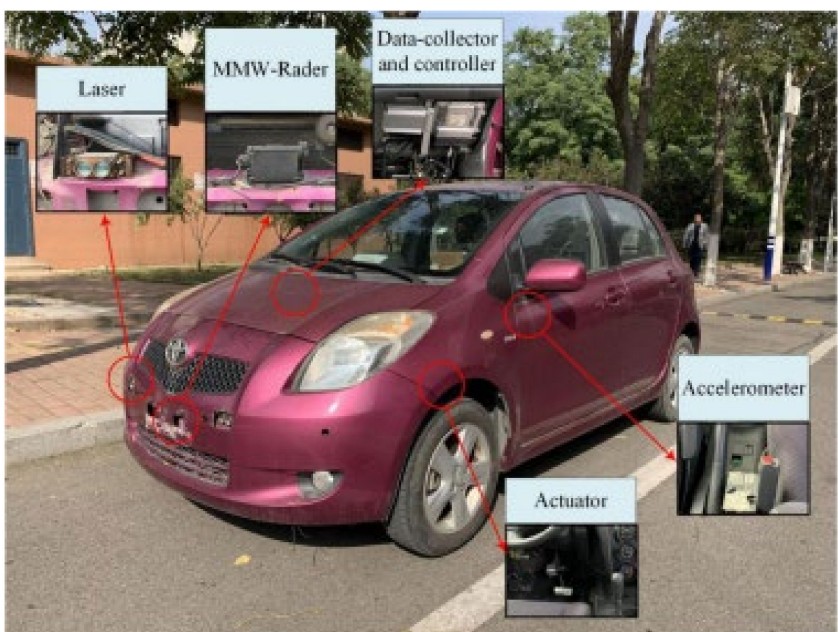

**Figure 3.** Configuration of the test vehicle.

For the first part, the speed of the front target vehicle was continually changed and the ACC system with APE was compared with the ACC system without APE (Case A). For the second part, two different scenarios were designed for the comparison of the ACC system with ADRC compensatory control and the ACC system without ADRC compensatory control. The change in the road slope and the weight of the host vehicle were used to change the internal and external disturbances. The first scenario was an uphill road with 5° gradient and 200 kg load (Case B), and the second scenario was a downhill road with a −5° gradient and 200 kg load (Case C).

### 3.1. Testing Results of Case A

At the beginning of this scenario, the distance between the host and the front vehicle was 5 m/s, and the initial speed of both vehicles was 0 m/s. First, the front vehicle accelerated from 0 s to 20 s. Then, the front vehicle decelerated from 25 s to 40 s. After the deceleration, the front vehicle accelerated again. In this test condition, the ACC systems with and without APE were compared to each other. Figure 4a shows that the ACC system with the APE method could accurately control the host vehicle tracking with less acceleration fluctuation, which resulted in better ride comfort. Figure 4a,b shows that ACC–APE can quickly and accurately trace the front target. The result of Figure 4c shows that ACC–APE had a smaller throttle angle than ACC without APE, resulting in a better fuel economy. The statistics of the acceleration results of the host vehicle are shown in Table 1. Compared with the ACC system without APE, ACC–APE has a lower acceleration average, standard deviation, and range.

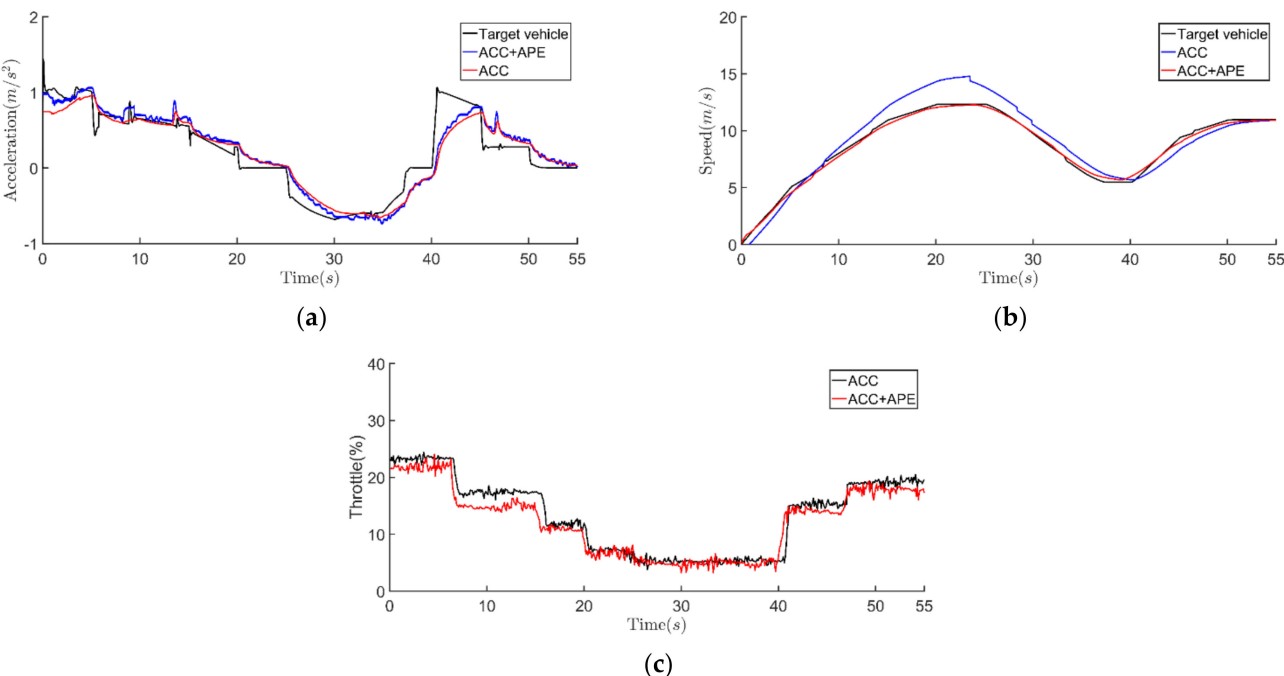

**Figure 4.** Comparison results for ACC and ACC−APE: (**a**) acceleration; (**b**) speed; (**c**) throttle.

**Table 1.** Statistics of the acceleration results of the host vehicle.

| Control Method | Average (m/s²) | Standard Deviation (m/s²) | Range (m/s²) |
|---|---|---|---|
| ACC | 0.2408 | 0.5188 | 1.8557 |
| ACC + APE | 0.2148 | 0.4688 | 1.6371 |

### 3.2. Testing Results of Case B

In this scenario, internal and external disturbances were added to the vehicle. The mass of the host vehicle was increased by an additional 200 kg and the test road is changed to a road with a 5° incline. Two controllers were compared: the ACC system with APE and ADRC (ACC–APE–ADRC); and the ACC system with APE (ACC–APE). The results from Figure 5a,b show that ACC–APE–ADRC accurately followed the front vehicle, whereas the ACC system without ADRC showed poor adaptability under the conditions of changes in the mass and the road. Figure 5d shows the compensation of the control input; the ADRC outputs combined with the output from the feed-forward controller were able to overcome the disturbances in the car-following mode. The statistics in Table 2 show that ACC with APE and the ADRC controller had a small mean speed error and lower standard deviation. This shows better tracking ability during the car-following process.

**Table 2.** Speed error statistics of the host vehicle with 200 kg load and 5° incline.

| Control Method | Average (m/s²) | Standard Deviation (m/s²) | Range (m/s²) |
|---|---|---|---|
| Without ADRC | 0.5656 | 0.9557 | 1.7469 |
| With ADRC | 0.2303 | 0.2819 | 0.9414 |

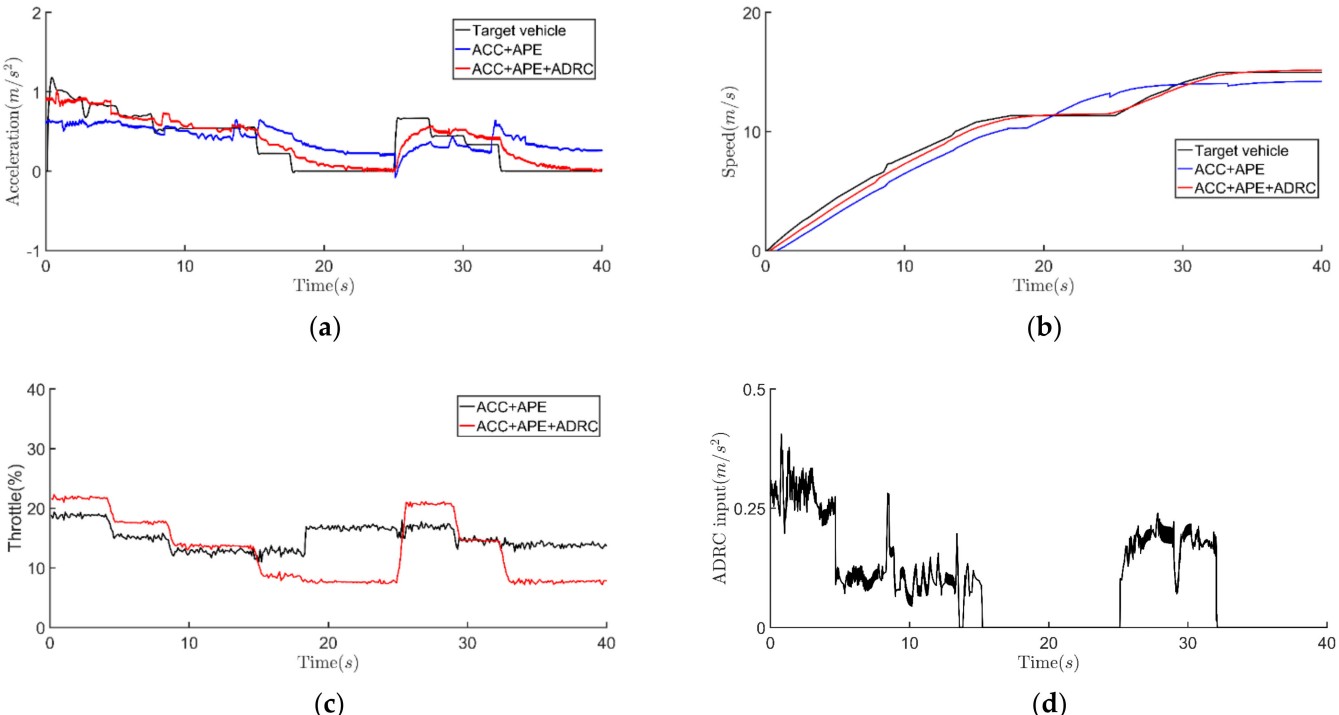

**Figure 5.** Comparison results for ACC−APE and ACC−APE−ADRC: (**a**) acceleration; (**b**) speed; (**c**) throttle; and (**d**) ADRC input.

### 3.3. Testing Results of Case C

In this test scenario, the internal and external disturbances were changed. The added mass of the host vehicle remained at 200 kg, but the test road was changed to one with a 5° decline. In this case, due to the road decline, the vehicle will accelerate more rapidly than in the case of a flat road. Thus, the host vehicle needs to reduce the throttle angle and increase the brake pressure when following the front vehicle compared to the flat road. ACC–APE–ADRC was compared with ACC–APE. From Figure 6a, due to the compensation calculated by ADRC, the host vehicle avoided a large acceleration in the 0 s to 20 s period and could brake appropriately. Figure 6b shows that the speed of the vehicle under ACC–APE was larger than that of the front vehicle during the tracing process, which may cause a rear collision. The input of Figure 6c shows that ADRC can compensate for both acceleration and deceleration. Table 3 shows that ACC–APE–ADRC can provide excellent tracking ability compared to ACC–APE.

**Table 3.** Speed error statistics of the host vehicle with 200 kg load and 5° decline.

| Control Method | Average (m/s$^2$) | Standard Deviation (m/s$^2$) | Range (m/s$^2$) |
|---|---|---|---|
| Without ADRC | 1.0247 | 1.4372 | 2.2415 |
| With ADRC | 0.1639 | 0.4654 | 0.8344 |

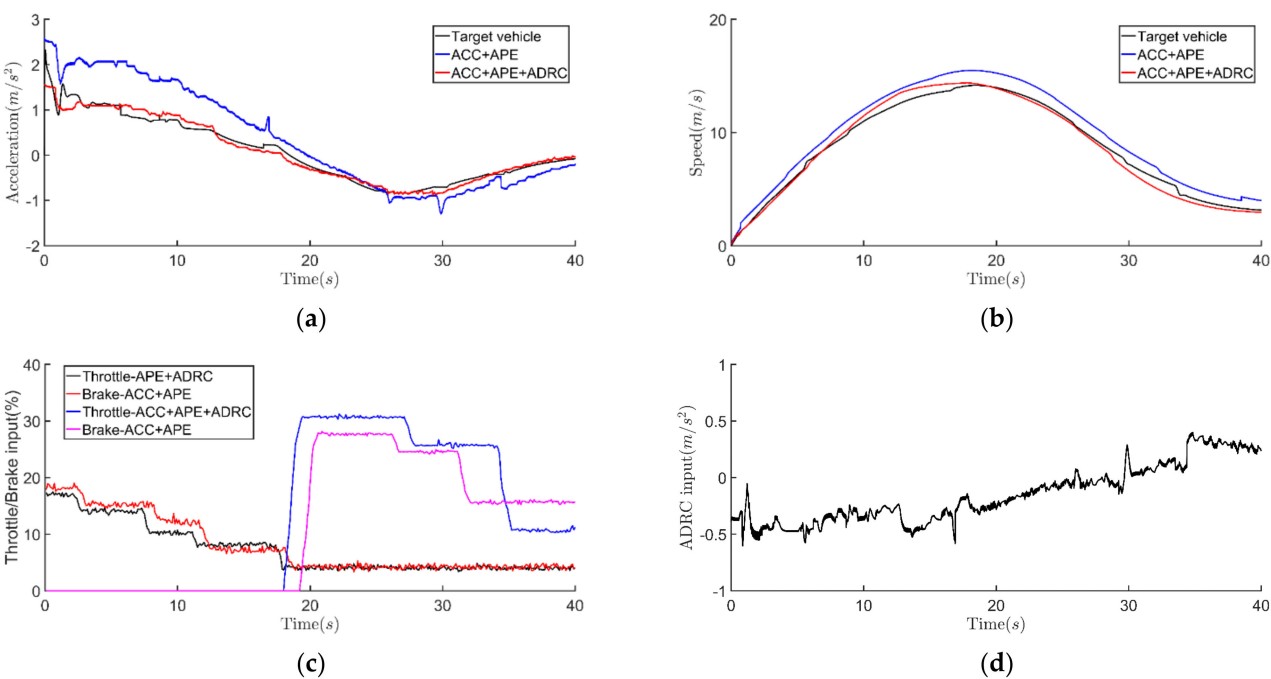

**Figure 6.** Comparison results for ACC−APE and ACC−APE−ADRC. (**a**) Acceleration; (**b**) speed; (**c**) throttle/brake input; and (**d**) ADRC input.

## 4. Conclusions

　　In this paper, an optimal design of the adaptive cruise control system based on MPC and ADRC compensatory control is proposed. To achieve the aim of enhancing safety, tracking capability, fuel economy, and ride comfort, the MPC method was introduced as the upper controller of the hierarchical construction. MPC can output an optimal command to the lower controller in each sample time period based on comprehensive considerations. However, it is unable to obtain an accurate solution if the prediction model is not correct; thus, an estimator of predictive acceleration was designed based on the least square method. Using this APE method in the MPC framework can increase control accuracy when the front target vehicle is accelerating or decelerating. After achieving the desired acceleration, the throttle or brake actuator is controlled to track the desired acceleration. Thus, acceleration feedback and compensatory control based on ADRC and VDM was used as the lower-level controller. This allows the host vehicle to accurately and safely follow the front target vehicle when subject to internal or external disturbances.

　　The proposed ACC–APE–ADRC controller and the ACC–APE controller were validated using road tests. Three different experimental cases were examined. The results of Case 1 showed that MPC with APE can perform better than MPC without APE in the ACC control system. The average, standard deviation, and range of the acceleration of ACC–APE were 10.80%, 9.64%, and 11.78% lower, respectively, than those of the ACC system without APE. The results of Cases 2 and 3 showed that feedback and compensatory control based on ADRC and VDM can overcome the influences of internal and external disturbances with lower speed error.

　　Based on the research results of this paper, two further areas of research are suggested; a complex and adaptable distance strategy should be used in the car—this remains for task for subsequent research following the model. In future research, a variable time headway strategy based on driving behavior will be introduced into the model. Second, the road tests in this paper only consider simple conditions. In the future, complicated and varying conditions will be considered to test the proposed controller.

**Author Contributions:** Conceptualization, Z.Y. and Z.W.; methodology, Z.Y.; software, Z.Y. and M.Y.; validation, Z.W.; formal analysis, M.Y.; investigation, Z.Y. and Z.W.; resources, Z.W.; data curation, M.Y.; writing—original draft preparation, Z.Y. and M.Y.; writing—review and editing, Z.Y., Z.W. and M.Y.; visualization, Z.Y. and M.Y.; supervision, Z.Y. and Z.W.; funding acquisition, Z.W. All authors have read and agreed to the published version of the manuscript.

**Funding:** This research was funded by the Shandong Provincial Natural Science Foundation, China, (grant numbers ZR2018 and MEE015).

**Institutional Review Board Statement:** Not applicable.

**Informed Consent Statement:** Not applicable.

**Data Availability Statement:** Not applicable.

**Conflicts of Interest:** The authors declare no conflict of interest.

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
