# Peer review of "An Optimization Design of Adaptive Cruise Control System Based on MPC and ADRC"

_actuators, doi:10.3390/act10060110_

Round 1

Reviewer 1 Report

The subject of the paper is interesting for the development of new control strategies for Adaptive Cruise Control  (ACC) Systems. To better achieve the function of the ACC system, a hierarchical control structure is  presented.

The paper is well written and interesting to read, however I see a couple of major issues that should be resolved before publishing this paper:

  • the meaning of the notations used in equation (1) is not explained (ak, KL, TL ...).
  • After equation (2) the authors state that v0 is the speed of the host vehicle. Please explain the meaning of the notation vk.
  • Please explain the equation (11). If k represents the current time what is the meaning of t? - Please explain the meaning of ap in equation (12) (in fact is af?)After equation (6) is stated that u is the control input; y is the control output. The expression is a bit ambiguous. Does y represent the output of the model (process)?
  • why is equation (8) not determined from the beginning in the form (20)?
  • from equation 24 it results that the MPC control is one step ahead MPC (prediction horizon = 1), which is not mentioned in the paper.
  • After Fig.(3) is stated 3 (d) shows that ACC-APE has smaller throttle angle than the ACC without APE, it means better fuel economy -  possible Fig.3 (c)  ?
  • For the results in fig. 3 ADRC is used? (for both cases - for ACC and ACC-APE?)

Reviewer 2 Report

I found the paper interesting and I congratulate the authors on the results of their work. The method and calculations seems to be correct and justified. Then you have implemented it and verified in an experiment. Anyhow, I suggest some minor amendmends:

  • in sec. 2.1 you write that "car" is not linear and time varing, and then "hence a first order" and you assume linear model. Maybe you should write "despite"?
  • a_f introduced in (4) is only explained in line 143. 
  • the index f for target vehicle (in my opinion) is a bit misleading as it can not distinguish a followed from follower car. You name the followed one a target vehicle, which is correct but ;-)
  • in (16) you have k->0 what does it mean?
  • in the sec 2.3 you present car dynamic models (2.3.1) and compensatory schemes (2.3.2) without any references. I guess these equations were not invented bu you.
  • experiment results - ACC is to keep the distance betwen two cars. I mean to compare and test these systems you should present distance errors, which would be more convincing.
  • There are also jerks that describe passangers comfort (you have some limits in (19)). So you could present the same analysis as for acceleration.
  • are tables 2 and 3 consisten with the text? should the avarage acceleration (error ?) with ADRC be bigger than without?

Round 2

Reviewer 3 Report

I'm satisfactory with the revised manuscript. However, it is strongly recommended to conduct the additional proofreading process with a native before proceeding the final publication process.
